# 1 **Aerosol extinction and backscatter Optimal Estimation retrieval for High**

## 2 **Spectral Resolution Lidar**

Sharon P. Burton<sup>1</sup>, Johnathan W. Hair<sup>1</sup>, Chris A. Hostetler<sup>1</sup>, Marta A. Fenn<sup>1,2</sup>, John A. Smith<sup>1</sup>, Richard A. Ferrare<sup>1</sup>

<sup>1</sup>NASA Langley Research Center, Hampton, VA, USA
<sup>2</sup>Coherent Applications, Inc., Hampton, VA, USA
Email: [Sharon.P.Burton@gmail.com](mailto:Sharon.P.Burton@gmail.com), [Johnathan.W.Hair@nasa.gov](mailto:Johnathan.W.Hair@nasa.gov)

### 8 **Abstract**

High Spectral Resolution Lidars (HSRLs) have been successfully deployed from a variety of platforms: ground based,  
airborne, and now satellite. These lidars are uniquely valuable for characterizing atmospheric aerosol and clouds,  
benefiting from the ability to characterize vertical variability in more detail than any passive instruments, and,  
compared to elastic backscatter lidars, provide additional channels of measurements that permit the direct retrieval of  
particulate extinction. Although analytic solutions exist for deriving particulate backscatter, extinction, and linear  
depolarization ratio, in the case of extinction, the analytic technique greatly magnifies measurement noise. Low signal-  
to-noise measurements stress the traditional inversion methods. Accordingly, algorithms for the retrieval of HSRL  
backscatter and extinction are re-examined and optimized to reduce the noise propagation. Here we explore an Optimal  
Estimation methodology and compare it with an implementation of the direct differentiation method like that  
historically used for the processing of airborne HSRL data from NASA Langley Research Center.

### 19 **1. Introduction**

Lidar instruments are routinely used for the study of aerosols in the atmosphere from the ground (e.g. Grund and  
Eloranta, 1991; Welton et al., 2001; Pappalardo et al., 2014, Jin et al. 2022), airborne platforms (e.g. Hair et al., 2008;  
Wirth et al., 2009; Carroll et al., 2022), and from space (Winker et al., 2009; Yorks et al., 2016; Flament et al., 2021;  
do Carmo et al., 2021; Ke et al., 2022), because of their ability to vertically profile atmospheric components. Lidars  
for the measurement of atmospheric aerosol are frequently polarization sensitive, and may be elastic backscatter lidar,  
Raman lidar or High Spectral Resolution Lidar (HSRL). The HSRL (Shipley et al., 1983) and Raman techniques  
(Ansmann et al., 1990) have the advantage over elastic backscatter in that one or more additional detector channels  
are employed to separate the backscattered light into components originating from aerosols or cloud particles versus  
air molecules, which in turn enables the retrieval of particulate (aerosol or cloud) extinction via an exact analytic  
solution of the lidar equation. This is a great advantage over elastic backscatter lidar, which requires added information  
from other instruments, climatology, or assumptions about the lidar ratio (that is, the extinction-to-backscatter ratio)  
to solve the lidar equation. Here we will focus on HSRL and, specifically, on the HSRL technique that has been
implemented for airborne lidar operated by NASA Langley Research Center (Hair et al., 2008; Burton et al., 2018).  
We limit the discussion to aerosol retrievals.
While purely analytic solutions for particulate backscatter and extinction coefficients and particulate linear  
depolarization ratio exist for HSRL and Raman lidar, these retrievals, as with all retrievals, are subject to measurement  
noise. For these systems, noise magnification is not an issue for the retrieval of particulate backscatter and linear  
depolarization ratio; however, the retrieval of extinction greatly magnifies the noise, because it involves the derivative  
of the log of the measurements. Therefore, the solutions of particulate extinction in circumstances with lower signal-  
to-noise levels can be challenging and sometimes unsatisfying. These circumstances can include relatively pristine  
aerosol environments with low signal, or situations where ~~the~~ measurement noise is large; the retrieval of aerosol  
extinction from space, for instance, presents a significant challenge even with an HSRL instrument (Donovan et al.,  
2020; 2024). This study discusses a retrieval from the HSRL measurement technique that is better suited to dealing  
with lower signal-to-noise, and which therefore can extend the usefulness of HSRL measurements into regimes where  
the traditional retrieval struggles.
In a theoretical noise-free retrieval of a number of unknowns from the same number of measurements, the analytic  
solution is the correct solution. However, if any noise is added, there is still only one exact solution but, in that case,  
it fits the signal plus the noise (i.e. overfitting) and therefore doesn't match the true noise-free signal. In cases like the  
extinction retrieval, the noise magnification potentially can significantly mask the true signal. Of course, the true  
signal is not known in general, except in the case of simulations. A common way to deal with noisy retrievals is to  
introduce regularization. The core idea of behind these strategies (Twomey, 1977; Tikhonov, 1977) is that a family of  
valid solutions exist that, when input into the forward model, can reproduce measurements, not exactly, but within  
measurement uncertainty. Regularization chooses among this family of solutions by introducing an additional  
requirement that the solution must be smooth in a particular defined way that depends on the type of regularization.  
The two requirements are represented in a cost function to be minimized.
Regularization has been used for Raman and HSRL retrievals in various ways (Shcherbakov, 2007; Pornsawad et al.,  
2008; Povey et al., 2014; Denevi et al., 2017; Garbarino et al., 2016; Marais et al., 2016; Donovan et al., 2020; Ehlers  
et al., 2022; Donovan et al., 2024). Shcherbakov (2007) first introduces regularization for a Raman lidar retrieval  
using a Tikhonov regularization with a second-difference smoothing matrix as the constraint, plus and smoothing of  
the measurement inputs. The variable to be solved for is the lidar ratio, rather than the extinction, because it is easier  
to constrain due to its more limited range of variability, a strategy we will continue to use in this study. Shcherbakov  
concludes that the Tikhonov regularization reduces the error over the conventional algorithm (the analytic solution  
with no regularization), but that the lidar ratio appears to be oversmoothed in his example. Pornsawad et al. (2012)  
follows up with a Levenberg-Marquardt method with a tunable regularization parameter, which does not require  
smoothing of the input measurements. Others (Garbarino et al., 2016; Ehlers et al., 2022) use Maximum Likelihood,  
which does not explicitly represent a constraint in a cost function, but instead they use more ad hoc methods of
constraint, either by ending the iterations early (Garbarino et al., 2016) or with a box constraint on the lidar ratio that  
forces it to stay within the defined bounds on each iteration (Ehlers et al., 2022). -These are practical methods but do  
not easily support an understanding of the impact of the constraint on the errors, or the quantification of uncertainty  
in general. Marais et al. 2016 adapt a Maximum Likelihood estimator from medical image processing called Total  
Variation Penalized Maximum Likelihood Estimator (TV-PMLE), which enforces piecewise smoothness both  
vertically and temporally. This method produces images with excellent noise reduction but again does not lend itself  
easily to the quantification of uncertainty.
This study focuses on a retrieval for particulateaerosol backscatter coefficient, linear depolarization ratio and lidar  
ratio from noisy HSRL signals, focusing specifically on aerosol. We useing Optimal Estimation (OE) (Rodgers, 2000),  
following other work using OE for Raman and HSRL (Povey et al., 2014; Donovan et al., 2024). Optimal estimation  
has the advantage of providing a framework that allows for propagation of all the error sources, not just random, but  
also calibration uncertainties and most importantly the uncertainty due to the constraint itself, which takes the form of  
a prior solution with specified uncertainty. Donovan et al. (2024) describe a combination retrieval for the spaceborne  
HSRL ATLID, where the state vector is the logarithm of aerosol extinction, lidar ratio, and effective radius to enforce  
positivity. In that work, the outputs of the analytic retrieval operating on coarse resolution are used as the “prior”  
information for the Optimal Estimation for defined atmospheric layers on a finer horizontal resolution. More usually,  
the prior is defined without using the measurements, as information that is available before the measurements are  
made. Povey et al. (2014) uses OE retrievals for Raman lidar using two different sets of state vectors, either extinction  
and backscatter, or the lidar ratio and the logarithm of the backscatter. In their setup, the prior is rather detailed and  
includes assumptions of the distribution of the aerosol concentration with height and an assumed correlation with  
height, as well as rather tight constraints on the lidar ratio, which are then loosened in a subsequent case study. With  
Optimal Estimation, whether using the standard Bayesian definition of a prior independent of measurements, or the  
more empirical approach of Donovan et al. (2024), the definition of uncertainties for the prior is crucial. If prior  
uncertainty truly reflects the degree of certainty in the prior, then there will be solutions that match both the prior and  
the measurements, within the respective uncertainties. However, if the prior and prior uncertainty are unrealistically  
restrictive, then there may not be any solutions that match the measurements.
Accordingly, this study uses a more conservative prior that represents minimal knowledge of the aerosol before the  
measurements are made. The HSRL technique, given the existence of the analytical solution, provides a strong  
constraint already and does not necessarily need much regularization. In this study, we achieve reasonable smoothing  
using a very loose and conservative prior uncertainty that does not imply foreknowledge of the aerosol type or lidar  
ratio, thereby minimizing required assumptions and potential biases associated with them. In addition, unlike several  
of the retrievals discussed above, except Povey et al. (2014), we formally include multiple potential sources of  
systematic measurement uncertainty and explore characteristics of the *a posteriori* uncertainty (that is, the solution  
uncertainty) including correlations in the solution vectors and the effective vertical resolution of the results. Section 2  
is dedicated to explaining the retrieval methodology, including the retrieval forward model, inputs and outputs, and  
3

uncertainty propagation. Section Error: Reference source not found presents results and discussions for two retrieval  
eases. Specifically, Section 3 discusses a simulated case, where knowledge of the exact truth allows for extensive  
validation of the retrieval and uncertainties, and the freedom to vary parameters enables quantitative discussion of the  
trade-off between uncertainty and resolution. Section 4 follows up with the application of the retrieval to a case of  
actual airborne measurements made by the HSRL-2 instrument. Section 5 concludes the discussion.

## 6 2. Retrieval Methodology

### 7 2.1. The forward model lidar equations

Light transmitted from the lidar is backscattered by particles and molecules in the atmosphere and is attenuated by the  
particles and molecules along the path between the lidar and the scattering volume, as in the following single-channel  
lidar equation.

$$P(r) = \frac{k}{r^2} \beta(r) T(r)^2 \quad (1)$$

where  $P(r)$  represents the lidar signal as a function of range  $r$ ; the variable  $\beta$  is the 180° backscatter coefficient, the  
scalar  \$kC\$  is a collection of range-independent instrument constants, and  $T(r)^2$  is a transmittance factor representing  
the reduction of light on the two-way journey between the lidar and the scattering volume at range  $r$ .

$$T(r)^2 = \exp \left\{ -2 \int_0^r [\alpha_{gas}(r') + \alpha_m(r') + S_p(r') \beta_p(r')] dr' \right\} \quad (2)$$

In Eq (2),  $\alpha(r)$  indicates a profile of extinction. The subscript *gas* refers to absorption from gaseous species, (e.g.  
ozone for the 532 nm lidar wavelength). The subscript *m* in these quantities refers to scattering from air molecules in  
the Rayleigh scattering regime, and *p* refers to particle scattering and absorption by aerosols and cloud particles. The  
particulate extinction is represented as the particulate 180° backscatter coefficient,  $\beta_p$ , multiplied by  $S_p$ , the lidar ratio  
(i.e., ratio of extinction to backscatter), following customary practice begun by Fernald et al. (1972). The two-way  
transmittance factor, when multiplied by particulate backscatter, gives the attenuated particulate backscatter, or when  
multiplied by the sum of the particulate and molecular backscatter gives the total attenuated backscatter.
While this retrieval is adaptable, we focus on the instrument configurations for NASA Langley airborne HSRL  
instruments: HSRL (Hair et al., 2008), HSRL-2 (Burton et al., 2018), HSRL-DIAL, and HALO (Carroll et al., 2022).  
See prior papers for discussions of direct retrievals, calibrations, and uncertainties (Hair et al., 2008; Burton et al.,  
2015; Burton et al., 2018). The particulate backscatter, extinction, and linear depolarization ratio retrievals are enabled  
by three main detector channels at a given wavelength. These signals are functions of the particulate and molecular  
attenuated backscatter. In the detector, the incoming light is split with a polarizing beam splitter into components that  
are parallel and perpendicular to the outgoing laser transmission. The parallel portion is further split to facilitate
separating the particulateaerosol backscatter from the molecular backscatter. (Hair et al. (2008) mentions a fourth  
 channel, a parallel total (particulateaerosol + molecular) channel, which simplifies some calculations, but since it is  
 not present in all versions of the instruments, we leave it out of this study). The signals are written in the following  
 general form.

$$\mathbf{P}_m(\mathbf{r}) = \frac{Kg_m}{g_p r^2} \left[ A \boldsymbol{\beta}_m(\mathbf{r}) \left\{ \frac{1}{2} - \chi \left( \frac{\delta_m}{\delta_m + 1} - \frac{1}{2} \right) \right\} + B \boldsymbol{\beta}_p(\mathbf{r}) \left\{ \frac{1}{2} - \chi \left( \frac{\delta(\mathbf{r})}{\delta(\mathbf{r}) + 1} - \frac{1}{2} \right) \right\} \right] T(\mathbf{r})^2 \quad (3)$$

$$\mathbf{P}_p(\mathbf{r}) = \frac{K}{r^2} \left[ C \boldsymbol{\beta}_m(\mathbf{r}) \left\{ \frac{1}{2} - \chi \left( \frac{\delta_m}{\delta_m + 1} - \frac{1}{2} \right) \right\} + D \boldsymbol{\beta}_p(\mathbf{r}) \left\{ \frac{1}{2} - \chi \left( \frac{\delta(\mathbf{r})}{\delta(\mathbf{r}) + 1} - \frac{1}{2} \right) \right\} \right] T(\mathbf{r})^2 \quad (4)$$

$$\mathbf{P}_\perp(\mathbf{r}) = \frac{Kg_\perp}{g_p r^2} \left[ \boldsymbol{\beta}_m(\mathbf{r}) \left\{ \frac{1}{2} + \chi \left( \frac{\delta_m}{\delta_m + 1} - \frac{1}{2} \right) \right\} + \boldsymbol{\beta}_p(\mathbf{r}) \left\{ \frac{1}{2} + \chi \left( \frac{\delta(\mathbf{r})}{\delta(\mathbf{r}) + 1} - \frac{1}{2} \right) \right\} \right] T(\mathbf{r})^2 \quad (5)$$

Where  $\mathbf{P}_\perp(\mathbf{r})$  indicates the range-dependent signal in the channel that receives the perpendicularly polarized portion,  
 and  $\mathbf{P}_m(\mathbf{r})$  and  $\mathbf{P}_p(\mathbf{r})$  indicate the molecular-dominated and particulate-dominated signals split from the parallel-  
 polarized portion, respectively. The scalar ratios  $g_m/g_p$  and  $g_\perp/g_p$  are relative calibration factors for the measurement  
 signals (henceforth called gain ratios), and the scalar  $K$  is an arbitrary scaling factor common to all three signals. Note  
 that the subscripts  $m$  and  $p$  are used for convenience in the channel signals and channel gain calibration constants,  
 even though the split between molecules and particulates is imperfect. The range-dependent particulate linear  
 depolarization ratio is  $\delta(\mathbf{r})$  while  $\delta_m$  represents molecular linear depolarization ratio, well approximated by the constant  
 value 0.0036 (Behrendt and Nakamura, 2002). The scalar parameter  $\chi$  allows for the potential for a small amount of  
 cross-talk between polarization channels due to polarization angle or ellipticity (Burton et al., 2015) and is unity in  
 the case of no cross-talk. The scalar parameters  $A, B, C, D$  indicate the split between molecular and particulate  
 components, which can be achieved in a few ways. For instance, for NASA Langley airborne HSRL instruments, the  
 HSRL technique at 532 nm is implemented by tuning the laser to an absorption feature of an iodine vapor filter in the  
 receiver. Doing so effectively removes the particulate backscattered light from the  $m$  channel, whereas there is no  
 filtering for the  $p$  channel. In that case  $B=0$  and  $C=D=1$  while  $A$  is the transmission of molecular scattering through  
 the iodine filter measured separately as in Hair et al. (2008) where it is called  $F$ . Alternately, an interferometer is used  
 for 355 nm for airborne HSRL and has been studied for potential use at 532 nm in a space-based instrument. In that  
 case, the molecular contribution is split evenly between the two channels by design of the interferometer, and the  
 particulate contribution is largely, but not completely, directed to the  $p$  channel. Therefore  $A=C=0.5$ , and  $B+D=1$   
 with  $B$  small and  $D$  large. The ratio  $D/B$  is a performance metric that we call the *contrast ratio* (Burton et al., 2018).  
 Multiple scattering is not included in these forward model equations. The current study focuses only on aerosol  
 scattering where multiple scattering is insignificant.
The numerical forward model is a discretized version of the lidar equations. Here we assume a grid of discrete slabs  
 having a uniform backscatter coefficient and lidar ratio throughout each slab. The range to the midpoint of slab  $i$  is  
 5

$\mathbf{r}(i)$  and the height is  $\Delta\mathbf{H}(i)$ . This formulation has the flexibility that the state vectors can be discretized at a coarser  
 resolution than the measurements, which aids in noise reduction. Povey et al. (2014) also solves for the state vector at  
 a coarser resolution than the measurements, but in that case by using cubic spline interpolation to translate between  
 grids. Since a splined state vector will not necessarily exactly reproduce the measurements even in a noise-free case,  
 we discretize the lidar equations as follows, with the transmittance due to passage of the light between the upper  
 boundary of slab  $i$  and its midpoint notated separately from the transmittance through all the overlying layers. This  
 separation makes explicit the dependencies on the current layer and on the overlying layers, enabling the computation  
 of the derivatives with respect to range-resolved quantities that are required for the OE retrieval.

$$\mathbf{P}_m(i) = \frac{g_m K'}{g_p \mathbf{r}(i)^2} \left[ A \boldsymbol{\beta}_m(i) \left\{ \frac{1}{2} - \chi \left( \frac{\delta_m}{\delta_m + 1} - \frac{1}{2} \right) \right\} + B \boldsymbol{\beta}_p(i) \left\{ \frac{1}{2} - \chi \left( \frac{\boldsymbol{\delta}(i)}{\boldsymbol{\delta}(i) + 1} - \frac{1}{2} \right) \right\} \right] \mathbf{T}(i)^2 \mathbf{T}(j > i)^2 \quad (6)$$

$$\mathbf{P}_p(i) = \frac{K'}{\mathbf{r}(i)^2} \left[ C \boldsymbol{\beta}_m(i) \left\{ \frac{1}{2} - \chi \left( \frac{\delta_m}{\delta_m + 1} - \frac{1}{2} \right) \right\} + D \boldsymbol{\beta}_p(i) \left\{ \frac{1}{2} - \chi \left( \frac{\boldsymbol{\delta}(i)}{\boldsymbol{\delta}(i) + 1} - \frac{1}{2} \right) \right\} \right] \mathbf{T}(i)^2 \mathbf{T}(j > i)^2 \quad (7)$$

$$\mathbf{P}_\perp(i) = \frac{g_\perp K'}{g_p \mathbf{r}(i)^2} \left[ \boldsymbol{\beta}_m(i) \left\{ \frac{1}{2} + \chi \left( \frac{\delta_m}{\delta_m + 1} - \frac{1}{2} \right) \right\} + \boldsymbol{\beta}_p(i) \left\{ \frac{1}{2} + \chi \left( \frac{\boldsymbol{\delta}(i)}{\boldsymbol{\delta}(i) + 1} - \frac{1}{2} \right) \right\} \right] \mathbf{T}(i)^2 \mathbf{T}(j > i)^2 \quad (8)$$

where

$$\mathbf{T}(i)^2 = \exp \left\{ - \left[ (\boldsymbol{\alpha}_{gas}(i) + \boldsymbol{\alpha}_m(i) + \mathbf{S}_p(i) \boldsymbol{\beta}_p(i)) \Delta\mathbf{H}(i) \right] \right\} \quad (9)$$

$$\mathbf{T}(j > i)^2 = \exp \left\{ -2 \sum_{j=i+1}^{TOA} \left[ (\boldsymbol{\alpha}_{gas}(j) + \boldsymbol{\alpha}_m(j) + \mathbf{S}_p(j) \boldsymbol{\beta}_p(j)) \Delta\mathbf{H}(j) \right] \right\} \quad (10)$$

The pre-multiplier  $K'$  accounts for molecular and gas transmittance between the laser and the top of the aerosol  
 solution grid (denoted TOA), as well as absolute scaling common to all channels.
Analytical solutions for particulateaerosol backscatter, extinction and linear depolarization ratio are discussed by Hair  
 et al. (2008) and Burton et al. (2018). While we will not repeat the equations for the analytic solution here, it is worth  
 pointing out that none of the solutions for particulateaerosol extinction, backscatter, or linear depolarization ratio  
 depend on the overall scaling factor  $K'$ , provided the channels are relatively calibrated (i.e. the gain ratios, contrast  
 ratio and depolarization cross-talk parameter are known). The backscatter and linear depolarization ratio are each  
 found from the ratio of channels (thus  $K'$  ratios out), while extinction is found from the derivative (with respect to  
 range) of the logarithm of the molecular-dominated channel (after correcting for particulate cross-talk), and the  
 derivative of a log also does not vary with any overall multiplier. In the OE retrieval,  $K'$  is solved as part of the state  
 vector as a nuisance parameter, since it is required to reconstruct the measurement vector that is used in the cost  
 function, but is expected to have little impact on the particulateaerosol backscatter, extinction, or linear depolarization
ratio. In our retrieval, we also solve for the depolarization cross-talk parameter  $\chi$ , while in the analytic method, this  
must be determined independently in a separate workflow by examination of any apparent depolarization in aerosol-  
free space (Burton et al., 2015).
The primary benefit of the retrieval in the current study is the reduced errors in the solutions for the lidar ratio and  
extinction profiles compared to the analytic retrieval, rather than the backscatter profile, which is already well behaved  
in the analytic solution even for relatively noisy situations. It is still a valid strategy to retrieve the particulate  
backscatter and depolarization ratio using the analytic solution on a fine grid spacing, and the lidar ratio (and thereby  
particulate extinction) at a coarser resolution using a regulated method (Shcherbakov, 2007). However, we present  
the full regularized retrieval of all parameters at the same grid spacing for completeness of the analysis.
**2.2. Optimal Estimation**
Optimal estimation finds the state vector which (when inserted into the forward model) optimally matches both the  
measurements and an *a priori* estimate of the state vector, both weighted by appropriate uncertainties. The quantity  
to be minimized includes the difference between the state vector and the prior, weighted by prior uncertainty, and the  
difference between the measurements and reconstructed measurements, weighted by measurement uncertainty, shown  
in the expression that follows:

$$\frac{[\mathbf{y} - \mathbf{F}(\mathbf{x})]^T \mathbf{S}_y^{-1} [\mathbf{y} - \mathbf{F}(\mathbf{x})] + [\mathbf{x} - \mathbf{x}_a]^T \mathbf{S}_a^{-1} [\mathbf{x} - \mathbf{x}_a]}{N} \quad (11)$$

Here  $\mathbf{y}$  represents the measurement vector and  $\mathbf{S}_y$  the measurement uncertainty covariance matrix. The retrieval vector  
is represented as  $\mathbf{x}$ , and  $\mathbf{F}(\mathbf{x})$  is the output of the lidar equations operating on  $\mathbf{x}$ . The prior profile  $\mathbf{x}_a$  is the best estimate  
of the state before measurements are taken, and  $\mathbf{S}_a$  is the prior covariance uncertainty matrix. The cost function is  
normalized by the total number of binned measurements in the three channels,  $N$ . The first term of this cost function,  
the measurement residual, is a quality indicator for the retrieval. Note that the actual error in each element,  \$\mathbf{y}-\mathbf{F}(\mathbf{x})\$ , is  
balanced by the uncertainties  \$\mathbf{S}_y\$ . Therefore, if the residual term is approximately one or less, that indicates the  
solution agrees well with the measurements within the measurement uncertainty.
The measurement vector  $\mathbf{y}$  comprises the signals in the three lidar detector channels, as a function of height in the  
atmosphere, after correcting the relative calibration and the range-squared dependence, i.e.,  $\mathbf{P}_m(r) * r^2 * g_p/g_m$ ,  $\mathbf{P}_p(r) * r^2$   
and  $\mathbf{P}_\perp(r) * r^2 * g_p/g_\perp$ . The three signal profiles are concatenated in the measurement vector  $\mathbf{y}$ . The measurements are  
cloud-screened and aggregated both horizontally and vertically. Horizontal aggregation can be relatively coarse and  
can be chosen to optimize the tradeoff between noise reduction and capturing true atmospheric variability. On the  
other hand, vertical aggregation of the measurements is limited to 15 m. Further noise reduction is achieved by making  
the solution's vertical grid coarser than the measurement grid, so that the number of inputs exceed the number of  
outputs. By using fine resolution measurements and coarsening the retrieval grid, the attenuation within each coarse
grid box is correctly calculated using the forward model during each updating step. This strategy avoids bias that  
would be caused by vertical averaging ignoring attenuation within the thickness of the bin.
The state vector to be retrieved contains profiles of the aerosol backscatter coefficient  $\beta_p(i)$ , the aerosol lidar ratio  
$S_p(i)$ , and the aerosol linear depolarization ratio  $\delta(i)$ , at all altitude levels on the coarser solution grid. The use of  
“aerosol” hereafter instead of “particulate” is meant as a reminder that we focus only on aerosol and have not included  
multiple scattering as would be required for solving for cloud properties. The overall scaling factor  $K'$  is also included  
in the state vector, although it is of no interest, since it is needed for mapping the state vector to the measurements.  
The depolarization cross-talk parameter is also included.
More familiar to a larger portion of the scientific community than lidar ratio or aerosol backscatter is aerosol  
extinction. The aerosol extinction profile is calculated from the state parameters by multiplying the aerosol backscatter  
and lidar ratio at each altitude. Aerosol extinction, like aerosol backscatter, scales with the amount of aerosol in an air  
parcel. The ratio of extinction to backscatter (lidar ratio), therefore, does not depend on the amount of aerosol present,  
but rather only on the properties of the aerosol particles including shape, size, composition, and humidification. Its  
values are generally between about 10 and 120 for all studied aerosol types at typical lidar wavelengths, whereas the  
extinction and backscatter can vary over orders of magnitude. For this reason, lidar ratio is easier to capture with a  
realistic prior than aerosol extinction is, which is the primary reason why lidar ratio, rather than extinction, is chosen  
for the retrieval state vector (Shcherbakov, 2007).
The numerical forward model is the discretized version of the lidar equations given in Eqs. (6)-(10). Other inputs to  
the forward model are the molecular number density profile and calibration parameters from Eqs. (3)-(5). The  
molecular number density profile is interpolated from the Modern-Era Retrospective Analysis for Research and  
Applications, version 2 (MERRA-2) (Buchard et al., 2017). The relative calibration gain ratios are not solved for in  
this retrieval but rather acquired in special calibration operations (Hair et al., 2008) and assumed to be known to within  
5%. This gain ratio uncertainty is included as a potential systematic uncertainty in the error budget. In addition, for  
the case we show here with an interferometer in the instrumentation, the contrast ratio must be determined using data  
from highly scattering targets such as clouds and is determined on a much coarser scale than the scale for the current  
retrieval (Burton et al., 2018). It is not included in the state vector but rather included as a known quantity with  
uncertainty of 5%. The cross-talk parameter  $\chi$  is relatively easy to retrieve even on a profile-by-profile basis and is  
therefore included in the state vector, where it is also easier to calculate its uncertainty.
Uncertainties are a critical input of the OE procedure. They are included for the calibration parameters and the  
measurement signals as a function of range. The OE framework handles uncertainties as full covariance matrices,  
which describe both the random and systematic features. An advantage of the framework is that the measurement and  
parameter errors can be easily written as uncorrelated errors in measurement space, and then straightforwardly
propagated into state space where they are more complex with both random and correlated (i.e. systematic)  
components.
From Rodgers (2000, Eq 2.27), we have an expression for the uncertainty covariance matrix for the retrieved state,  $\mathbf{\hat{S}}$ ,  
given the uncertainty covariance matrix for the measurements and the prior.

$$\mathbf{\hat{S}} = (\mathbf{J}^T \mathbf{S}_y^{-1} \mathbf{J} + \mathbf{S}_a^{-1})^{-1} \quad (12)$$

Here,  $\mathbf{J}$  is the Jacobian matrix of partial derivatives of the signals with respect to the state parameters,  $\partial \mathbf{y} / \partial \mathbf{x}$ . These  
derivatives are calculated analytically from Eqs (6)-(10).  $\mathbf{S}_a$  is the variance-covariance matrix of prior uncertainties.  
$\mathbf{S}_y$  is the variance-covariance matrix constructed from both the random measurement uncertainty and systematic  
uncertainties from the calibration parameters, using

$$\mathbf{S}_y = \mathbf{S}_r + \mathbf{J}_b \mathbf{S}_b \mathbf{J}_b^T \quad (13)$$

where  $\mathbf{S}_r$  is the diagonal matrix of random variances for the signals.  $\mathbf{S}_b$  includes the uncertainties in the gain ratios and  
contrast ratio, and  $\mathbf{J}_b$  is the matrix of derivatives of the signals with respect to these calibration parameters, derived  
from Eqs (6)-(10).
The measurement uncertainties are shown in the sections describing the simulated case and real data case below.  
Calibration uncertainties are taken to be 5% for the gain ratios and for the contrast ratio ( $D/B$ ), consistent with  
discussions in HSRL-2HSRL2 instrument papers (Burton et al., 2015; Burton et al., 2018). While the measurement  
errors are assumed to be entirely random and the calibration uncertainties reflect a (potential) constant bias error, the  
propagated errors (in the state vector) might take more complex forms. For instance, an error in the relative calibration  
between the particulateaerosol and molecular channels will directly correspond to a bias in the derived  
particulateaerosol backscatter, but an error in the contrast ratio (cross-talk parameters) has a very non-linear systematic  
effect in the lidar ratio, producing errors primarily at the edges of layers (Burton et al., 2018).
A prior is included for regularization, but we choose to specify large prior uncertainties with no correlation and  
furthermore we make sure that the resulting OE solution agrees with the measurements within the stated uncertainties.  
Accordingly, the prior uncertainty covariance matrix represents no assumed knowledge of the aerosol type, only of  
the overall range of values that lidar ratio can take. Specifically, the prior profile of backscatter is taken to be zero  
with a one-sigma standard deviation of 0.015 km<sup>-1</sup> sr<sup>-1</sup>, and the lidar ratio is taken to be 50 sr with a one-sigma standard  
deviation of 35 sr, (i.e. 95% confidence the lidar ratio falls between -20 sr and 120 sr for a normal distribution) with  
zeros on the off-diagonals (i.e. no correlation between levels at the coarse resolution of the solution). Specifically, the  
prior profile of lidar ratio is taken to be 50 sr with a one-sigma standard deviation of 35 sr, (i.e. 95% confidence the  
lidar ratio falls between -20 sr and 120 sr for a normal distribution). For aerosol backscatter, the prior is taken to be  
zero with a one-sigma standard deviation of 0.015 km<sup>-1</sup> sr<sup>-1</sup>. Aerosol backscatter can vary over many orders of
magnitude. This standard deviation covers a large portion of the range of values seen in many years of airborne  
 HSRL-2 data. Of course, lidar ratio and aerosol backscatter are not distributed normally, but this setting for prior  
 uncertainty is large enough that the shape of the distribution is unimportant. Although there can be no negative aerosol  
 backscatter values, choosing zero as the prior is helpful since the prior will come into play primarily when the  
 measurement signal is insufficient to constrain the results; that is, when aerosol backscatter is near zero. Since the  
 standard deviation is large and the prior is therefore relatively weak, it does not bias results when the measurement  
 signal-to-noise ratio is larger. Finally, the prior covariance matrix has zeros on the off-diagonals (i.e. no correlation  
 between levels at the coarse resolution of the solution). Of course, lidar ratio and particulate backscatter are not  
 distributed normally, but this uncertainty is large enough that the shape of the distribution is unimportant.
The cost function is minimized using the stepper equation given by (Rodgers, 2000) as Eqn 5.8. For the cases  
 presented here, the cost function is minimized in about 4 steps.
A further calculation produces the uncertainty covariance matrix for aerosol extinction via propagation of uncertainty  
 for derived quantities (see also Knobelspiesse et al., 2012; Burton et al., 2015),

$$\mathbf{S}_{\text{ext}} = \left( \left( \frac{\partial \boldsymbol{\alpha}}{\partial \mathbf{x}} \right)^T \mathbf{S} \frac{\partial \boldsymbol{\alpha}}{\partial \mathbf{x}} \right)^{-1} \quad (14)$$


Where the partial derivative of extinction with respect to backscatter at the same altitude is the lidar ratio, and the  
 partial derivative of extinction with respect to lidar ratio at the same altitude is the backscatter. Derivatives of  
 extinction with respect to state quantities at other levels are zero.
**Figure 1. Simulation of smoke over marine aerosol,  
 with a small amount of dust in between.**

1    3. Simulated spaceborne HSRL case

2    3.1. Construction of simulated case

3    The first test case consists of simulated profiles of known aerosol backscatter coefficient, lidar ratio and linear  
4    depolarization ratio profiles and known error characteristics. This test is carried out to assess errors due to the retrieval  
5    methodology itself, along with the measurement and instrument errors consistent with a notional spaceborne HSRL  
6    with assumptions given in Table 1. This simulation does not assess additional errors due to sub-grid atmospheric  
7    variability. Figure 1 illustrates simulated test profiles of aerosol backscatter, lidar ratio, linear depolarization ratio, and  
8    extinction. The shape and magnitude of the profiles are based on data products from airborne HSRL-2HSRL2  
9    measurements made during the ORACLES field mission (Redemann et al., 2021; Burton et al., 2018; Harshvardhan  
10   et al., 2022) with some additional variability added to the aerosol properties for testing. The aerosol backscatter  
11   coefficients, lidar ratios, and particulateaerosol linear depolarization ratios are defined on vertical coarse grid of 285  
12   m spacing, which is the grid that will be used for retrieving these profiles. Since the grids are the same, it is  
13   theoretically possible for a retrieval to exactly reproduce the truth profile; therefore, any differences with the truth  
14   profile are indicative of retrieval error and propagated measurement and instrument error, as distinct from sub-grid  
15   atmospheric variability.

16   **Table 1. Instrument parameters for a notional space based HSRL used for generating realistic simulated uncertainties.**

| Lidar parameter             | Value  |
|-----------------------------|--------|
| Laser pulse energy          | 100 mJ |
| Laser repetition rate       | 70 Hz  |
| Receiver transmittance      | 50%    |
| Telescope diameter          | 100 cm |
| Lidar (orbit) altitude      | 450 km |
| Photon detection efficiency | 13%    |
| Excess noise factor         | 1.4    |

Figure 2. Symbols show the simulated range-square-corrected relatively calibrated lidar signals in three detector channels, using the assumed instrument parameters for a notional space-based instrument given in Table 1, with signal averaging of 15 m vertically and 50 km horizontally. From left to right, greenblue diamonds indicate the perpendicular channel multiplied by a factor of 10; orangepink circles show the molecular-dominated backscatter channel, and black squares show the particle-dominated channel. All channels are range-square corrected and relatively calibrated. The overall calibration factor is set to 1.

The simulated truth state parameters are then placed on a 15-m grid ranging up to 12 km. Eqs. (6)-(10) are applied to generate the noise-free signals on that finer grid. A realistic molecular density profile is selected (not shown) and used to compute molecular backscatter and extinction at 355 nm. The contrast ratio is set to 35, and the relative calibration ratios are set to unity. To complete the simulation, Gaussian noise is added to the signals using assumed instrument parameters from Table 1 for a notional space-based instrument with signal averaging of 15 m vertically and 50 km horizontally. The simulated noisy signals are illustrated in Figure 2. (The perpendicular channel measurements are multiplied by a factor of 10 to be seen on the same scale.) Uncertainties of 5% for the gain ratios and contrast ratio are included in the retrieval framework and propagated into the retrieval uncertainty covariance matrix, although no actual error is added to these parameters in the simulation. The molecular profile from MERRA-2 is assumed to be known with no uncertainty.

**Figure 3.** Retrieval results are shown for the simulated case introduced in Figure 1. BlueGreen is used for the Optimal Estimation results and orange is used for the analytical inversion. Error bars represent the propagated standard deviation uncertainty, including random and systematic uncertainty for OE and random measurement uncertainty only for the analytic solution.


## 2 3.2. Retrieval results

Figure 3 shows the primary output of the retrieval, the profiles of the particulateaerosol backscatter coefficient, lidar  
 ratio and aerosolparticulate linear depolarization ratio. The retrieval grid is 285 m compared to 15 m for the simulated  
 measurements. Both the results from Optimal Estimation and the results from the analytic method applied at the same  
 285 m vertical grid spacing are illustrated. The particulateaerosol extinction is also shown for convenience, calculated  
 as the product of particulateaerosol backscatter and lidar ratio. Profiles of the propagated standard deviation  
 uncertainty (from the diagonal of the *a posteriori* covariance matrix) are also shown as error bars. For the analytic  
 retrieval uncertainties in these plots, only random measurement uncertainty is propagated, making these somewhat  
 underestimated. In contrast, the illustrated OE uncertainties include systematic uncertainties as discussed above. The  
 retrieved calibration constant is  $0.99 \pm 0.01$  and the retrieved value of  $\chi$  is  $1.001 \pm 0.002$  (compared to a true simulated  
 value of 1.0 for each).
For the lidar ratio and extinction, the OE solution is smoother than the analytical solution and has a smaller uncertainty.  
 This is the benefit of the optimization and shows that the method addresses our primary concern with the analytic  
 method, the unruly propagation of error in the extinction retrieval. For particulateaerosol backscatter and  
 particulateaerosol linear depolarization ratio, the analytic method is well behaved and has never given any concern.  
 The good agreement between the two methods for those quantities is a good check on both methods.

**Figure 4.** The differences between the simulation and retrieval results for (a) aerosol backscatter as a percentage, (b) lidar ratio as a percentage, (c) aerosol linear depolarization ratio as an absolute difference, and (d) aerosol extinction as a percentage are shown as thick bluegreen lines with symbols (orange for the analytical inversion), while the thin lines show the envelope defined by the *a posteriori* uncertainty.


Figure 4 shows the propagated uncertainty as thin lines on the negative and positive side of zero, along with the actual  
 error (retrieval minus truth) shown as thick lines with symbols. For aerosol backscatter, extinction and lidar ratio,  
 these are given as percent differences (with truth as the denominator), and as absolute differences for aerosol linear  
 depolarization ratio. The solutions are unbiased, varying around 0, within the theoretical propagated uncertainty  
 envelopes.

Figure 5. These figures show measurement residual profiles in bluegreen for OE and in orange for the analytic retrieval for (a) the molecular dominated attenuated backscatter channel, (b) the particulate dominated attenuated backscatter channel, and (c) the perpendicular attenuated backscatter channel. The residuals are the difference between the simulated measurements and the reconstructed measurements. The measurement uncertainty envelopes are represented in black.

Inserting the retrieved profiles of aerosolparticulate backscatter, lidar ratio, and linear depolarization ratio into the  
 forward model produces reconstructed measurement vectors, which provide another check on the OE solution. Figure  
 shows the profiles of residuals: the differences between the forward-modelled retrieval results and the simulated  
 measurements. Since the OE retrieval reproduces the state vector very well, the profiles of measurements reconstructed  
 from the OE solution agree very well with the noise-free simulated truth measurements. The differences with the noisy  
 simulated measurements are therefore very similar to the simulated noise itself and are consistent with the  
 measurement uncertainty shown. The normalized residual for this case is 0.95, distributed evenly among the three  
 channels, and the total cost function (including the penalty for disagreement with the prior) is 0.96.
We note that the analytic method does not reproduce the shape of the measurement profiles with as much skill,  
 although it too produces variability very similar in magnitude to the simulated noise. This difference is because the  
 OE solution involves a global minimization, while the analytic method is purely local, solving for a state vector at a  
 given altitude considering only the measurements close to the same altitude. Attenuated backscatter, which is  
 proportional to the measurement vector, is not local, since it depends sing on the extinction at every altitude above it.  
 Therefore, even though the residual is small at the highest altitude, random errors in the derived state contribute to  
 accumulating errors in the attenuated backscatter at lower altitudes. These do not call into question the solutions for  
 backscatter and extinction, which were already shown to be unbiased in Figure 4.
Error bars and uncertainty envelopes in Figure 3Figure 4 and Figure 4 reflect only the diagonal elements of the state  
 error covariance matrix. It's useful to check potential cross-correlation of error by looking at off-diagonal elements.
Specific blocks of the correlation matrix are illustrated in Figure 6. The top panels in Figure 6 show the uncertainty  
correlations for state variables of the same type at different altitudes, with, of course, 100% correlation on the diagonal.  
We see that the uncertainty in the particulateaerosol backscatter has significant correlation throughout the profile. This  
cross-correlation is related to the uncertainties in the relative calibration (gain ratios) and contrast ratio which were  
included as potential systematic error sources. Recall that the uncertainty in retrieved aerosol backscatter is small; this  
correlation shows that much of that small uncertainty is due to the systematic uncertainty. Likewise, off-diagonal  
correlation in the particulateaerosol linear depolarization ratio uncertainty is primarily related to depolarization cross-  
talk parameter  $\chi$ , but again the overall uncertainty is low, and that indicates that the small uncertainty is dominated by  
systematic uncertainty. The pattern in the lidar ratio uncertainty covariance, with some negative correlations on the  
off-diagonals nearest to the diagonal, reflect the fact that the lidar ratio is related to the difference between nearby  
measurement bins, and also reflects the impact of the contrast ratio. There is little or no correlation between  
uncertainties in the lidar ratio at distant altitudes, since the lidar ratio depends only on these nearby differences and  
has no dependence on absolute scaling. The bottom panels of Figure 6 show cross-correlations between uncertainties  
for state parameters of different types. There is some cross-correlation between the particulateaerosol backscatter and  
lidar ratio uncertainties at the same level (i.e. along the diagonal) which is related to the cross-talk between the  
particulate-dominatedaerosol and molecular-dominated channels, reflecting that an error in the contrast ratio causes a  
bias in retrieved aerosolparticulate backscatter and oscillation in the lidar ratio (Burton et al., 2018). There is little  
cross-correlation in uncertainty between aerosol linear depolarization ratio and either of the other profile quantities,  
except in isolated strips at high altitude where there is no aerosol and so very little signal.
Figure 6 shows correlation between profile quantities, but the covariance matrix also includes rows that indicate Error: Reference source not found shows the correlation between the uncertainty in the profile quantities and the scalar quantities, overall scaling parameter  \$K'\$  and the three state variable vectors (backscatter, lidar ratio, and depolarization eoneatenated) as well as the retrieved polarization cross-talk parameter  \$\chi\$ . There is significantLarge negative correlation (not shown) withbetween the uncertainty in overall scaling factor,  \$K'\$ , and the full backscatter uncertainty profile again reflects that any errors in the backscatter profile are correlated with the calibration constant, as expected.  
Correlations of the overall scaling factor uncertainty with lidar ratio and aerosol depolarization ratio uncertainties are near zero. Likewise, there is predictably significant correlation between the uncertainty in the depolarization cross-talk parameter, Error: Reference source not found likewise shows the correlation in the uncertainty of the vectors with the scalar parameter  \$\chi\$ , the depolarization cross-talk parameter, which is predictably high for correlation in uncertainties between  \$\chi\$  and the aerosolparticulate depolarization ratio uncertainties, whereas correlations between uncertainties in this parameter and the aerosol backscatter and lidar ratio are near zero. All of these patterns are expected and reflect that errors (although small) in the backscatter profile are partially systematic and correlated with the calibration constant and likewise that the errors in the depolarization ratio profile are partially systematic and correlated with the depolarization cross-talk parameter.

**Figure 6.** Portions of the *a posteriori* error covariance matrix are shown which illustrate the correlation in propagated uncertainty between the particulateaerosol backscatter at all altitudes (top-left), the lidar ratio at all altitudes (top-middle), and the aerosolparticulate linear depolarization ratio at all altitudes (top-right), as well as cross correlations between particulateaerosol lidar ratio and backscatter (bottom-left), particulateaerosol linear depolarization ratio and backscatter coefficient (bottom-middle), and aerosolparticulate linear depolarization ratio and lidar ratio (bottom-right).


**3.3. Degrees of Freedom and Effective Resolution**
The averaging kernel matrix is given by

$$\mathbf{A} = (\mathbf{J}^T \mathbf{S}_y^{-1} \mathbf{J} + \mathbf{S}_a^{-1})^{-1} \mathbf{J}^T \mathbf{S}_y^{-1} \mathbf{J} \quad (15)$$

and is closely related to the propagated variance-covariance matrix  $\hat{\mathbf{S}}$  (compare Eq (12)).
The trace of the averaging kernel matrix is the degrees of freedom (DOF) of the signal (Rodgers, 2000), a measure of
how much information the signal (that is, the measurements) contributes to the retrieval of each quantity. Profiles of
the degrees of freedom of the signal are shown in Figure 7(a). The calculation for extinction uses a similar transform  
 as in Eqn (14) and relates to the variance-covariance matrix  $\mathbf{S}_{\text{ext}}$  instead of  $\mathbf{S}$ . For aerosol backscatter coefficient and  
 linear depolarization ratio, the DOF is near unity for the whole profile, since even in the presence of noise, the analytic  
 solution is hardly improved by the additional consideration of the prior. For the lidar ratio (and by extension, for the  
 extinction which is formed from the lidar ratio and backscatter), the DOF varies with altitude, being largest in regions  
 with the largest signals. The inverse of the DOF gives an estimator of the effective vertical resolution of the retrieval,  
 since it shows the scale over which elements of the solution vector are independent. The effective resolution varies  
 between quantities and with altitude, as shown in Figure 7(b).
In this study we distinguish between the “grid spacing” and the “effective resolution”, where effective resolution refers  
 to the inverse of the DOF. The effective resolution can be quantitatively interpreted as an estimate of the distance  
 over which elements of the retrieved state vector can be considered independent and uncorrelated. For instance, in a  
 region of the profile with high signal-to-noise ratio, it may be possible to retrieve the same or nearly the same number  
 of independent state vector elements as measurements, whereas in regions with lower signal, more neighboring  
 measurements must be considered to produce a single independent estimator of a state vector element. For aerosol  
 backscatter and linear depolarization ratio, the effective resolution is essentially constant and very close to the grid  
 spacing, reflecting the nearly perfect information content of the measurements. For aerosol lidar ratio and extinction,  
 the effective resolution is coarser than the solution grid, especially in regions of the profile where the signal is smaller  
 and the measurement uncertainty is correspondingly greater. In this case, there is some active regularization of the  
 profile due to the use of the prior, and this reduces-coarsens the effective resolution beyond the grid scale.

**Figure 7. (a)** Profiles of the degrees of freedom of the signal for each retrieved quantity (plus aerosol extinction) as a function of altitude, for the simulated case discussed above. **(b)** The inverse of the DOF is a measure of the effective vertical resolution of the retrieval.


### 21 3.4. Trade-off between uncertainty and effective resolution

The solution grid being coarser than the measurement grid is important for noise reduction. Multiple measurements  
 fall within a single vertical bin, reducing the impact of measurement noise. For the solution of aerosol backscatter
coefficient, the propagated uncertainty drops roughly proportional to the square root of either the grid spacing or the  
 effective vertical resolution, as more measurement points are incorporated into each vertical bin. This is illustrated in  
 Figure 8(a) which shows the propagation of random uncertainties for the OE and analytic retrievals of backscatter in  
 the altitude range from approximately 2.5 to 5 km as the retrieval was repeated multiple times at different grid scales.  
 (For this figure, systematic errors in calibration parameters were ignored in the error propagation for both retrieval  
 methods).

Figure 8. As the spacing of the vertical solution grid is varied in the simulated data case, the propagated uncertainty and effective vertical resolution of the aerosol backscatter coefficient retrieved in the 2.5-5 km altitude range are shown in (a). (b) shows the aerosol extinction uncertainty and the vertical resolution of the extinction and lidar ratio. In both panels, black indicates the results for both the Optimal Estimation and blue-green indicates the analytical retrieval. The x-coordinate of each data point is the effective vertical resolution, while thin black lines connect back to the numerical value of the grid scale on the same axis. In the case of aerosol backscatter coefficient in (a), the grid and resolution are very similar, so the lines are mostly vertical.

 Error: Reference source not found Figure 8(b) illustrates the effective vertical resolution and uncertainty for the aerosol  
 extinction coefficient, at around 2.5-5 km altitude, from both retrievals. The y-axis of this figure again represents the  
 propagated random uncertainty, neglecting systematic uncertainty in calibration and cross-talk parameters. Since the  
 analytic solution uses no information other than the signals (i.e. no prior), the effective vertical resolution is simply  
 the grid scale; that is, the retrieval errors at each grid point are independent of those at other grid points. However,  
 for the Optimal Estimation retrieval, the effective vertical resolution is coarser than the vertical grid. The x-axis  
 location of the black dots indicates the effective vertical resolution, as defined in the previous section. Thin black  
 lines connecting to the axis also indicate the grid scale used for each retrieval; for example, using a vertical grid scale  
 of 405 m led to an effective vertical resolution of 473 m. The propagated uncertainty from the analytic solution of  
 extinction falls off rapidly as the grid is coarsened, more rapidly than the backscatter curve seen above, and the  
 effective vertical resolution for the analytic solution is the same as the grid scale.
For OE, there is a minimum achievable effective resolution for extinction. This reflects the balance between the  
measurement uncertainty term and the prior uncertainty term in the cost function. Retrievals at finer grid scales don't  
achieve finer effective vertical resolution because the prior has a greater impact; the measurement uncertainty at these  
fine grid scales asymptotes to the prior uncertainty. and the resolution is coarse because solutions in neighboring grid  
points are not independent of the prior solution or of each other. At grid scales coarser than the one that leads to the  
minimum effective resolution, the usual trade-off between propagated uncertainty and vertical resolution prevails,  
with coarser resolutions associated with smaller propagated uncertainties. The grid scale used for the solutions shown  
in Section 3 was 285 m, very close to the scale that produces the minimum effective vertical resolution. To achieve a  
similar uncertainty, the analytic retrieval would have to be performed at a grid scale or resolution coarser than 500 m.  
For any grid scale, the extinction uncertainty from the OE solution is smaller than that in the analytic retrieval.
**4. Real data case**
Finally, we apply the retrieval to a real measurement case, using observations at 355 nm from the airborne HSRL-2HSRL2. These observations were made near the end of a transit flight to the ORACLES field mission, 26 August  
2016. This measurement was located approximately 460 km west of Walvis Bay, Namibia. The relatively calibrated  
measurement signals are shown in Figure 9. The reason for selecting this case rather than one from the core ORACLES  
science mission is the presence of dust, which exercises the ability to solve for aerosol linear depolarization ratio. The  
specific profile was also selected because it is cloud free. Further development of the optimal estimation retrieval is  
required to work with cloudy profiles, but that is beyond the scope of this initial study.  

**Figure 9.** Symbols show measured range-square-corrected relatively calibrated lidar signals in three detector channels for the real data case. From left to right, bluegreen diamonds indicate the perpendicular channel multiplied by a factor of 10; orange circles show the molecular-dominated backscatter channel, and black squares show the particulate-dominated channel.


The two gain ratios required for Eqs (3)-(5) as well as the filter function are determined as in Hair et al. (2008). These  
 quantities are already applied to the input of the algorithm; these are therefore relatively calibrated signals. The  
 uncertainties in the gain ratios for this retrieval are taken to be 5%. No absolute calibration is given or assessed for  
 HSRL-2HSRL2 data, since the analytic solution does not require it. Within the optimal estimation algorithm, it's  
 included as a nuisance parameter to be retrieved, merely to match the overall scaling within the cost function.
At 355 nm, the HSRL-2HSRL2 instrument makes use of an interferometer with very good but not perfect separation  
 between the two HSRL channels. For this study, data from three 355 nm channels were used: total (particulateaerosol  
 plus molecular) perpendicular signal, molecular dominated parallel signal and particulateaerosol dominated parallel  
 signal.
Details of the NASA Langley's HSRL-2HSRL2 measurement strategy at 355 nm as well as the algorithm used to  
 determine the contrast ratio (sometimes called cross-talk) between the two HSRL channels are given by Burton et al.  
 (2018). The molecular signal is split between the two channels evenly by design, so  $A=C=0.5$  in Eqs (3)-(4), and
there is negligible light lost within the interferometer so  $B+D = 1$ . In the selected case, the estimated contrast ratio  
( $D/B$ ) is 57.35 with an uncertainty of 5%. Thus,  $B = 0.017$  and  $D = 0.983$  in Eqs (3)-(4).
Standard archival products from HSRL-2HSRL2 for the ORACLES field mission use a 60-second average for the  
retrieval of extinction, and a 10-second average for backscatter. Here for this example, a 10-second averaged signal  
profile was used to test the retrievals in the presence of more noise. The vertical range of the data is from the ocean  
surface to 12 km. The data were binned onto a 15-m vertical grid, after screening data near and below the ocean  
surface.
Random uncertainties estimated from instrument parameters are provided in some but not all archived HSRL-2HSRL2  
files. Instead of these, we estimate the measurement uncertainties using the local vertical standard deviation over a  
running 150-m window for use in Eqn (11). This may artificially inflate uncertainties near the edges of aerosol features  
but is sufficient for the tests presented in this paper.

## 12 4.1. Results

Even with horizontal averaging minimized, there is less noise than the simulated case, so we are able to solve with a  
finer vertical grid spacing of 165 m. Figure 10 shows the analysis of the behavior of the uncertainty and effective  
vertical resolution for the extinction profile as the grid spacing is changed, similar to Figure 8(b). This analysis is  
performed between 1.5 and 2 km, which is near the maximum in the retrieved aerosol backscatter profile (Figure 14).  
This figure shows that for this case, the finest effective vertical resolution for the extinction (and lidar ratio) profile  
that can be achieved is about 220 m, which occurs for the retrieval with 165 m grid spacing.
**Figure 10.** The extinction uncertainty and  
effective vertical resolution from an Optimal  
Estimation retrieval on the real data case are  
shown for the altitude range 1.5-2 km, as the  
vertical grid for the retrieval is varied. Thin  
lines connect each point to the value of the grid  
spacing for that run on the x-axis scale.
Figure 11 shows the results for the OE and analytic retrievals of the real data case, for the altitude range where there  
 is measurable aerosol. As in the simulated case, the retrieval results for aerosol backscatter and linear depolarization  
 ratio are very similar between the two retrievals, while the retrieved lidar ratio and extinction are less noisy from OE  
 than from the analytic retrieval and likewise have smaller derived uncertainties. We note that the displayed analytic  
 retrieval of aerosol extinction is noisier than the publicly available HSRL-2HSRL2 archived product because the 10  
 second average is less horizontal smoothing than used in standard processing.
The overall scaling factor is solved with an uncertainty of 0.13%, which is a much smaller uncertainty than the  
 simulated case, due to the smaller measurement noise and finer vertical grid spacing, leading to, in effect, more  
 independent measurements of the aerosol-free part of the profile. The solution for the depolarization cross-talk  
 parameter is  $0.9917 \pm 0.0007$ .

**Figure 11.** A real data case from measurements made by airborne HSRL-2 near the end of a transit flight to the ORACLES field mission, 26 August 2016. This measurement was located approximately 460 km west of Walvis Bay, Namibia. The profile was selected because it is cloud free and because the enhanced depolarization due to the presence of some dust in a layer below 1 km altitude tests the ability to solve for aerosol depolarization ratio. (a)-(d) show the retrieved state variable profiles from the optimal estimation (bluegreen) and the traditional analytic method (orange). (e)-(h) show the a posteriori uncertainties of the OE retrieval and the propagated random uncertainty for the analytic retrievals; which these are also represented as error bars on the top panels. The relative uncertainties are calculated using the retrieved value in the denominator.


The correlations for the uncertainties for the profiles of aerosol backscatter, lidar ratio and depolarization ratio from  
 the Optimal Estimation retrieval are illustrated in blocks as before in Figure 12, for the entire altitude range of the  
 retrieval. As before, large correlations indicate that the uncertainties are dominated by systematic uncertainty, and  
 frequently occur where the total uncertainty is relatively small. Error: Reference source not found illustrates the  
 23

correlations in uncertainties between the overall scaling factor  $K'$  and the profile quantities, while Error: Reference  
source not found illustrates the correlations between the uncertainties in the depolarization cross talk parameter  $\chi$   
with the uncertainties in profile quantities. The correlation matrix for the aerosolparticulate backscatter is  
strongly diagonal, with some enhanced correlation at higher altitudes where there is little aerosol, due to the increasing  
influence of the prior. There is less correlation otherwise since the uncertainty in the overall scaling is very low. Some  
off-diagonal correlation in the aerosolparticulate linear depolarization ratio throughout the profile is again also related  
to uncertainty in the depolarization cross-talk parameter. There is some cross-correlation between the  
aerosolparticulate backscatter and lidar ratio uncertainties close to the diagonal which is related to the cross-talk  
between the aerosolparticulate and molecular channels, reflecting that an error in the contrast ratio causes a bias in  
retrieved aerosolparticulate backscatter and oscillation in the lidar ratio (Burton et al., 2018). Again, there is very little  
cross-correlation in uncertainty between aerosol linear depolarization ratio and either of the other profile quantities.  
The rows of the correlation matrix corresponding to the scalar quantities are similar to the simulated case, again  
reflecting the expected sources for the systematic uncertainties. Specifically, As expected, the largest correlations exist  
between uncertainties for with the overall scaling factor are forwith the backscatter profile uncertainties, and between  
for the depolarization cross-talk parameter with the aerosolparticulate depolarization ratio profile, with near zero  
correlations for other combinations.

Figure 12. Like Figure 6 but for the real data case, these panels illustrate the correlation in propagated uncertainty between the particulateaerosol backscatter at all altitudes (top-left), the lidar ratio at all altitudes (top-middle), and the aerosolparticulate linear depolarization ratio at all altitudes (top-right), as well as cross correlations between particulateaerosol lidar ratio and backscatter (bottom-left), aerosolparticulate linear depolarization ratio and backscatter coefficient (bottom-middle), and aerosolparticulate linear depolarization ratio and lidar ratio (bottom-right).


The profiles of residuals illustrated in Figure 13 demonstrate that the retrieval is of good quality and reproduces the  
 measurements within the measurement uncertainty, and therefore did not suffer undue influence from the prior profile.  
 The cost function is 1.02 with a total normalized residual of 0.99. Separated by channel, the normalized residuals are  
 0.84 for the molecular-dominated attenuated backscatter channel, 1.10 for the particulateaerosol-dominated channel,  
 and 1.05 for the perpendicular channel.

**Figure 13. Measurement residual profiles in bluegreen for OE and in orange for the analytic retrieval for (a) the molecular dominated attenuated backscatter channel, (b) the particle dominated attenuated backscatter channel, and (c) the perpendicular attenuated backscatter channel.**

The effective resolution derived from the degrees of freedom of the measurements is shown in Figure 14. As in the simulated case, the vertical grid scale (here 165 m) is the minimum possible value of the effective resolution derived from the degrees of freedom. The aerosol backscatter coefficient profile meets this minimum throughout the profile, and the aerosol linear depolarization ratio profile meets the minimum through a large portion of the profile. The lidar ratio (and extinction, which is derived from it) have coarser effective resolution, which varies with altitude. Their minimum values occur where the signals are strongest and therefore the information content is the greatest, but then above 4 km, the effective resolution becomes very large as particulateaerosol loading approaches zero and the prior profile comes into effect. The effective vertical resolution of lidar ratio and extinction are also large at 0.5 km, where aerosol loading is also small and the prior has significant impact.

**Figure 14.** Profiles of effective vertical resolution of the OE retrieval of aerosol backscatter (blue circles), lidar ratio (orange squares), aerosol linear depolarization ratio (pink triangles), and extinction (black pluses), defined in the information-content sense described above. The vertical grid scale is 165 m.


## 2 5. Conclusion

An Optimal Estimation procedure has been constructed for the retrieval of aerosol backscatter, lidar ratio and linear  
 depolarization ratio from noisy High Spectral Resolution Lidar measurements. Simulated data is retrieved, matching  
 both the simulated true state and the simulated measurements with high accuracy. The retrievals of lidar ratio and  
 extinction have smaller errors and smaller uncertainties than analytic retrievals with the same data set. Residuals are  
 examined and are in excellent agreement with the measurements within measurement uncertainty.
The particular strengths of the OE regularization technique are that it adapts to the effective amount of information at  
 each level, avoiding over-smoothing when the SNR is large while at the same time limiting unruly propagation of  
 error when the SNR is small. In this study, we avoid over-constraining the solution by using a loose and conservative  
 prior uncertainty. This is feasible due to the relatively high information content of HSRL measurements, along with  
 the use of a solution grid that is coarser than the measurement grid, permitting the same benefits as smoothing of the  
 inputs, but with a formally correct accounting of the attenuation within each layer of the coarser grid. In this study we  
 illustrate the trade-offs between effective resolution and uncertainty using our methodology and demonstrate how the  
 effective resolution and degrees of freedom vary throughout the state profile.
Another benefit of OE compared to other regularization methodologies is the relative ease of mathematically  
 propagating both random and systematic uncertainty, including the systematic uncertainty related to the assumed prior  
 27

solution itself. In this study, we are careful to include other important potential sources of systematic error, including  
relative calibration and cross-talk calibration parameters, and include the overall calibration as a nuisance parameter.  
While it is easy to characterize input uncertainties as either random (e.g. measurement noise) or systematic (e.g.  
calibration errors), after propagation through the lidar equations, output uncertainties are not as easy to describe in  
those simple terms. For example, contrast ratio or cross-talk calibration error can lead to greater errors at sharp  
gradients in the profile, and uncertainties can be correlated within a profile or between quantities. Covariance matrices  
are shown to provide the complete characterization of propagated uncertainty.
Finally, the retrieval is also demonstrated here for a real data case of airborne HSRL-2HSRL2 data from the  
ORACLES field mission, implementing a finer horizontal resolution than was used for processing the field mission  
archive data. Future work on automated data preparation, including cloud clearing, would allow the practical use of  
this retrieval for full flight curtains. For application to space or for solving within optically thin clouds or the tops of  
clouds would also require treatment of multiple scattering, as in Donovan et al. (2024) or Mason et al. (2023).

## 13 **6. Data and code availability**

The HSRL-2 ORACLES data set is archived at doi: 10.5067/SUBORBITAL/ORACLES/ER2/2016\_V1. Code used  
to create the results in this study is available on request from Johnathan.W.Hair@NASA.gov.

## 16 **7. Author contribution**

SPB: conceived the study, formed the algorithm, created retrieval code, performed the retrievals, and analyzed results.  
JAS and CAH provided modelling of the hypothetical satellite instrument. JWH and CAH are the creators of the  
airborne instrument and derived the calibration procedures and original retrieval strategies that inform the current  
work. MAF processed and maintains the HSRL data that were used in this study. SPB prepared the manuscript with  
contributions from all authors.

## 22 **8. Competing interests**

The authors declare they have no conflict of interest.

## 24 **9. Acknowledgments**

Color tables used in this work include a continuous version of the Purple-Orange diverging color scheme of Cynthia  
Brewer at Color Brewer 2.0 (<https://colorbrewer2.org>). For discrete colors, Brewer colors as implemented in the  
Coyote library for IDL and Discrete Batlow (Cramer et al., 2020) was used. We thank Fabio Cramer and Cynthia  
Brewer, Cynthia Brewer, and David Fanning for providing these color tables and the implementation.
**10. Financial support**
This work was funded by the National Aeronautics and Space Administration (NASA) Earth Science Division.

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
