# Peer review of "Aerosol extinction and backscatter Optimal Estimation retrieval for High"

_EGUsphere, 2025_

## Author Comment (AC1)

Response to Reviewer 1
Thank you for your positive comments and your suggestions for improving the manuscript. Your time and effort are appreciated. Responses to your individual suggestions are given below in blue.

Reviewer 1 states:

The article describes an algorithm for an improved extinction retrieval from high spectral resolution lidar (HSRL) measurements. An analytical solution exists for the extinction coefficient derived from HSRL measurements, which often suffers from low signal to noise ratios. The presented optimal estimation solution has the benefit of less noisy extinction coefficient and lidar ratio profiles, which is desirable for aerosol research with lidar. Similar approaches were already reported for Raman lidar measurements and spaceborne HSRL observations. Here, the approach is developed for the airborne HSRL of NASA Langley Research Center. Another highlight is the characterisation of measurement uncertainties. The article is clearly written and describes the algorithm and applies it to simulated and real measurement data. The topic fits in the scope of AMT and should be published after minor revisions which mostly concern the figures.

Minor comments:

1.      While the article focusses on the application to airborne HSRL systems, the possibility of the application to ground-based HSR lidars should be considered. New ground-based HSRL systems were developed, e.g., Jin et al., 2020. Jin, Y.; Nishizawa, T.; Sugimoto, N.; Takakura, S.; Aoki, M.; Ishii, S.; Yamazaki, A.; Kudo, R.; Yumimoto, K.; Sato, K. & Okamoto, H.: Demonstration of aerosol profile measurement with a dual-wavelength high-spectral-resolution lidar using a scanning interferometer, Appl. Opt., Optica Publishing Group, 2022, 61, 3523-3532
**Several of the included references are papers about ground-based lidars, and the one you suggest is also a good addition. It has been added in the revision.**

2.      P4L20: Here, you call the lidar system HSRL-2, later just HSRL2. It is up to you to decide how to name your lidar system.
**In the revision, this is made more consistent, with HSRL-2 used instead of HSRL2 in all cases.**

3.      Eq 11 is not really an equation (no =). Furthermore, the journal standards expect a vector to be in bold italic.
**It's true that it's an expression and not an equation. I don't think that's disallowed by journal standards, and I somewhat prefer not to introduce a new symbol for the left-hand side that isn't required elsewhere in the text. As for the typesetting of variables, in the revision, the formatting of scalar, vector and matrix variables has been made more consistent with journal standards.**

4.      In nearly all figures, the units of the optical properties are missing.
**Units have been added to Figures 1-3, 6, 12, and 14-15 for the revision.**

5.    Furthermore, the figures are often quite small. Sometimes, the minor ticks are not resolved well.
The overcrowded minor ticks were removed. These were part of multi-panel figures that I don't think are practical to split, since other reviewers are already concerned that there are too many figures. I hope the publisher's layout will allow the figures to be a little larger than how I set them in the review copy.

6.    Fig 2 + 12: The perpendicular signal (in green) is multiplied by a factor of 10, not the particle signal. The label inside the figure is wrong.
True, that was a mistake.  It's been fixed in the revision.

7.    Fig 2: There are no pink circles as mentioned in the caption.
Fair enough, pink is not a very good descriptor word for this color.  The colors were chosen to be distinguishable to viewers with color-vision deficiencies as well as viewers with typical vision, but the minor tradeoff is that they are not well described by the most common color words.  The caption is revised to use "orange" as a better description for this color.

8.    You use the terms "particle/particulate" and "aerosol" synonymously in the description of the optical properties and signals, e.g., particulate depolarization ratio and aerosol depolarization ratio or particle(-dominated) signal or aerosol signal. It is ok, if you focus on aerosol and exclude clouds. However, it would be nice to harmonize the terms throughout the manuscript and especially the axis labels in the figures. Sometimes, the particle dominated channel is referred to as "para" – parallel in the axis labels (e.g., Fig 5+18).

In the revision, more care has been taken to use the labels aerosol and particulate consistently.  At the beginning of the manuscript, the descriptions of the measurement signals are general as to whether they involve aerosol or cloud, so "particulate" is used in the discussion almost exclusively at first.  However, when it comes to describing our particular retrieval, it seems important to remind readers that we are focusing only on aerosol, since solving for cloud particles would require a solution for multiple scattering that we have not included. Therefore, where we begin to describe specifics of the methodology, we make the vocabulary more specific and replace "particulate" with "aerosol" when referring to backscatter, extinction, and lidar ratio quantities, while keeping "particulate" when referring to measurement channels. To make the intent more clear, this sentence was also added at the start of the discussion of the solution vector:
*The use of "aerosol" hereafter instead of "particulate" is meant as a reminder that we focus only on aerosol and have not included multiple scattering as would be required for solving for cloud properties.*

"Para" was removed from Figures 5 and 18. The captions use the full descriptive text for clarity.

9.    The green lines in all your figures look almost blue to me. Maybe you find a different type of green to be clear.
The color has been changed to be better visible to people with red-green color vision deficiency and now the color is even more blue, so the description word has also been changed from green to blue in all the captions.

10.    P12L16-17 Why the systematic uncertainties are not included in the analytic retrieval?

The ability to propagate systematic uncertainties in a complete and consistent manner is one of the strengths of this new retrieval.  Although there have been published accounts describing and analyzing sources of systematic uncertainties for the analytic retrieval (e.g. Burton et al. 2015 and Burton et al. 2018), propagating them through the analytic retrieval has never been part of the standard processing.

11.    P14L13-14: Probably, this is the reason why the analytic retrieval has the lowest residual in the topmost kilometer for all three signals.
By "this" do you mean the statement at lines 13-14?  Yes, you are correct.  The residual is small at the top of the profile and the accumulation effect means the error is larger below that.  I added a few words to incorporate your idea.  The sentence now says "Therefore, even though the residual is small at the highest altitude, random errors in the derived state contribute to accumulating errors in the attenuated backscatter at lower altitudes."

12. P14L17 Fig 1 does not contain uncertainty estimates. Probably, you want to refer to a different figure here. ?
You are correct.  It should have been Figure 3.  It has been changed in the revision.

13.    Fig 6 + 15: I would add a, b, c … to the subplots.
   Added

14.    Please estimate how important are Fig 7+8 and Fig 16 + 17. Maybe a description in the text is sufficient. It is up to you to decide.
OK, given that two reviewers thought there were too many figures, this suggestion of which figures to replace with text is a good solution.  That has been done in the revision.  Figures 7 and  8 are replaced with this paragraph:
   *Figure 6 shows correlation between profile quantities, but the covariance matrix also includes rows that indicate correlation between the uncertainty in the profile quantities and the scalar quantities, K' and χ. There is significant negative correlation (not shown) between the uncertainty in the overall scaling factor, K', and the backscatter uncertainty profile, as expected. Correlations of the overall scaling factor uncertainty with lidar ratio and aerosol depolarization ratio uncertainties are near zero. Likewise, there is predictably significant correlation between the uncertainty in the depolarization cross-talk parameter, χ, and the aerosol depolarization ratio uncertainties, whereas correlations between uncertainties in this parameter and the aerosol backscatter and lidar ratio are near zero.  All of these patterns are expected and reflect that errors (although small) in the backscatter profile are partially systematic and correlated with the calibration constant and likewise that the errors in the depolarization ratio profile are partially systematic and correlated with the depolarization cross-talk parameter.*
Figures 16 and 17 are replaced with similar but shorter summary text.

15.    May I suggest, furthermore, to combine Fig 10 and 11 as Fig 10a and 10b?
OK, they have been combined in the revision

16.    Fig 18: The exponents 10^x are hard to read.
These have been made bigger.

17.     Please add a section about the code availability.
**The following statement has been added: Code used to create the results in this study is available on request from Johnathan.W.Hair@NASA.gov.**

---

## Author Comment (AC2)

Response to Reviewer 2.

Thank you for your positive and constructive comments. Your time and interest are greatly appreciated. Answers to your specific comments are given below in blue.

Reviewer 2 states:

I should start by saying, that this is an excellent and deep research. Calculating the the aerosol extinction coefficient is a challenging task, particularly in the presence of noise. The authors convincingly demonstrate that the Optimal Estimation (OE) technique reduces uncertainty in the calculations compared to traditional analytical methods. Additionally, the approach enables the determination of the optimal height resolution for different altitude ranges. Another key strength is its ability to account for both random and systematic uncertainties, including calibration errors.

Overall, this is a high-quality scientific manuscript that is certainly publishable in its current form. However, while certain aspects may seem self-evident to the authors, readers less familiar with OE may benefit from additional explanations.

Technical comments

1. Eq.4. Symbol "C" is the same as in Eq.1?
No, it isn't. Thanks for catching that. It's been changed in Eqn 1 to a unique symbol, lower-case $k$.

2. p.7 ln.18 "If the residual is approximately one or less, then the solution agrees well…". Would be good to explain or provide a reference.
In the revision, more explanation is added, like this: "Note that the actual error in each element, $y$-F($x$), is balanced by the uncertainties $\mathbf{S_y}$. Therefore, if the residual term is approximately one or less, that indicates the solution agrees well with the measurements, within the measurement uncertainty."

3. p.9 ln 19. "the prior profile of backscatter is taken to be zero". Probably needs explanation, why it is zero. Just wonder, if a prior profile can be calculated from standard analytical approach.
More explanation is added to the revision. It now reads like this:
> *Specifically, the prior profile of lidar ratio is taken to be 50 sr with a one-sigma standard deviation of 35 sr, (i.e. 95% confidence the lidar ratio falls between -20 sr and 120 sr for a normal distribution). For aerosol backscatter, the prior is taken to be zero with a one-sigma standard deviation of 0.015 $km^{-1}$ $sr^{-1}$. Aerosol backscatter can vary over many orders of magnitude. This standard deviation covers a large portion of the range of values seen in many years of airborne HSRL-2 data. Of course, lidar ratio and aerosol backscatter are not distributed normally, but this setting for prior uncertainty is large enough that the shape of the distribution is unimportant. Although there can be no negative aerosol backscatter values, choosing zero as the prior is helpful since the prior will come into play primarily when the measurement signal is insufficient to constrain the results; that is, when aerosol bacskcatter is near zero. Since the standard deviation is large and the prior is therefore relatively weak, it does not bias results when the measurement signal -to-noise ratio is larger. Finally, the prior covariance matrix has zeros on the off-diagonals (i.e. no correlation between levels at the coarse resolution of the solution).*

The idea of calculating a prior from the analytic approach is something that some authors have used. It's important to distinguish between a prior and first guess; they may or may not be the same state vector, but they play different roles. The first guess is used to quickly start the iterations in a regime that is believed to be close to the correct answer; therefore, the analytic solution makes a very good first guess. However, the role of the prior is to encode any information that exists *outside* of the measurements (literally "prior" to the measurements), so using an alternate inversion of the same measurements doesn't satisfy that role. To explore the idea further, the prior uncertainty is what's critically important, even more than the prior state vector. If the prior covariance matrix correctly reflects the amount of knowledge outside the measurements (what you would know if you made no measurements), then even if the prior state vector is "borrowed" from the analytic retrieval, that state vector will only be perceptible in the OE results where the measurement signal is relatively weak, which are the same locations where the analytic retrieval is most unstable.  So, I don't think there would be significant benefit to using the analytic retrieval for the prior state vector (but again, there is clear benefit to using it for the first guess). If you consider using the *uncertainty* for the analytic solution as a prior uncertainty, that would be a definite handicap, since the balance between the prior uncertainty and the measurement uncertainty is what drives OE to find the best smooth solution that fits the measurements.  It would also disable the ability to calculate DOF of the measurements and the associated effective resolution.

4. Fig.5. What are the units? Are measurements normalized?
The figure labeling for Figs 2,5, 12, and 18 has been revised to say "arbitrary units".  These values are the measurements or simulated measurements after relative calibration (i.e. multiplying by the gain ratios) and removing the range-squared dependence by multiplying by $r^2$. The figure annotations have also been changed to make this more explicit using the same symbols as in the equations, e.g. the quantity obtained for the molecular channel is $P_m r^2 g_p/g_m$.  The question was specifically about Fig 5. Fig 5 has the same units as Fig 2, since it is just a difference (at each specific grid point), and likewise for Fig 18 and Fig 12.  The overall scale is different between the simulation and the observed measurements, because the overall calibration constant is different.  In the case of the simulation, since the overall scaling is 1, this quantity can be thought of as the attenuated backscatter.

5. Fig.11. I have difficulty to understand this figure. Why effective resolution in OE method increases in  non-monotonic way? For grid of 50 m the uncertainty is the highest, but effective resolution is also high (700 m). Would be good to provide more explanations.
The effective resolution is not monotonic because there is a tradeoff or balance between the prior and the measurements in the cost function.  At the finest grid scales, the retrieval pushes the limit of the information content of the measurements, and the prior begins to take effect.  At 50 m, the prior plays a big role.  The uncertainty is large because the prior uncertainty is large, and the effective resolution is also coarse because solutions at neighboring grid points are not independent (since they are all highly dependent on the prior information).  This is the explanation given in the revision:

> For OE, there is a minimum achievable effective resolution for extinction. This reflects the balance between the measurement uncertainty term and the prior uncertainty term in the cost function.  Retrievals at finer grid scales don't achieve finer effective vertical resolution because the prior has a greater impact; the measurement uncertainty at these fine grid scales asymptotes to the prior uncertainty, and the resolution is coarse because solutions in neighboring grid points are not independent of the prior solution or of each other.  At grid scales coarser than the one that leads to the minimum effective resolution, the usual trade-off between propagated uncertainty and vertical resolution prevails, with coarser resolutions associated with smaller propagated uncertainties.

6. Fig.14f. At altitude of ~0.5 km uncertainty of OE is higher than for analytical solution. Can it be explained?

This single data point is within the clear air gap between two aerosol layers. In that region, the measurement information content is small and the uncertainty for both methods is quite large. I suspect the reason that the OE uncertainty is larger in Figure 14f at that point is that systematic uncertainties were very carefully included in the OE error analysis in this study, whereas the analytic uncertainties shown here are only the propagated measurement noise. This is mentioned elsewhere in the text, but in the revision, the figure captions for Figures 3 and 14 are also revised to reflect this. For example, the caption to Figure 14 (now Figure 11) now includes "(e)-(h) show the a posteriori uncertainty of the OE retrieval and the propagated random uncertainty for the analytic retrieval; these are also represented as error bars on the top panels."

---

## Author Comment (AC3)

Response to Reviewer 3.

Thank you for your careful reading and valuable comments. We appreciate the time and effort you put into reading and improving our paper. Responses to specific comments appear below in blue.

Reviewer 3 writes:

> Review of "Aerosol extinction and backscatter optimal estimation retrieval for high spectral resolution lidar" by S. P. Burton, J. W. Hair, C. A. Hostetler, M. A. Fenn, J. A. Smith, and R. A. Ferrare, proposed for publication in Atmospheric Measurement Techniques.

In this article, the authors present a method based on optimal estimation to retrieve lidar-based optical parameters (particulate backscatter, extinction, lidar ratio, depolarization ratio) in aerosols based on HSRL measurements in 3 channels (cross-polarized, and co-polarized Mie-dominated and Rayleigh-dominated). They compare results from that approach to those obtained using the more widely used analytical approach. They consider a theoretical case study, and actual measurements from the NASA Langley airborne HSRL instrument in a scene that mixes different aerosol layers. They discuss the advantages of the optimal estimation approach for each retrieval parameter, and introduce some of its peculiarities, like the effective vertical resolution.

The subject matter is new, interesting and valuable for lidar experts, especially now that HSRL measurements from space are widely available. The writing is clear and effective. The methodology is well-supported by appropriate references. I appreciate that the authors took the time to remind readers of the basics, and enlightening comments can be found throughout the manuscript. Figures are well-designed, most convey a clear and useful message with evidence that moves the discussion forward. The results show that the method proposed by the authors brings significant improvements to retrievals of aerosol extinction compared to the analytical approach. Even though the paper could very well be published as-is, I have minor comments that mainly hope to make the paper slightly more approachable to non-specialists.

Minor Comments

1. Figs. 1, 2, 3, 6, 7, 8, 12, 14, 15, 16, 17 : please add units for parameters when relevant (backscatter, lidar ratio, extinction, altitude)
Units have been added to Figures 1-3, 6, 12, and 14-15 for the revision. Figures 7,8, 16, and 17 have been removed, as suggested below and by another reviewer.

1. p. 6, L.15: "the backscatter and depolarization are each found from the ratio of channels" - how is the backscatter obtained from the ratio of channels? I could not find how this aligns with eq 4 in Hair 2008 or any equation in Burton 2018. As I understand it the backscatter is directly proportional either to the Pm (for molecular backscatter) or Pp (for particulate backscatter) channels, no ratio here.
Yes, aerosol backscatter is proportional to $P_p$ and molecular backscatter is proportional to $P_m$, but in both cases, there is the unknown molecular attenuation

T². In the HSRL method, the two channels are accurately calibrated relative to each other (i.e. the ratio $g_m/g_p$ is known), so taking the ratio of the two channels allows the T² factor to drop out, leaving the ratio of aerosol to molecular backscatter (plus one). The molecular backscatter is furthermore known (with small uncertainty) from an externally provided profile of molecular number density, so the aerosol backscatter can be solved from there simply by subtracting one from the ratio and multiplying that by the molecular backscatter. Equation 3 in Burton et al 2018 shows the simplest form of this. Equation 4 in Hair et al 2008 appears a little bit more complicated because all three channels are considered (including the perpendicular channel that allows for solving for depolarization), but the idea is the same.

2. Section 3 is huge (17 pages out of 28, not counting the references). Splitting 3.1 and 3.2 into their own sections could help make things more balanced.

Good suggestion and easy to implement. This has been done in the revision.

3. The authors introduce in Section 3.1.3 the "effective resolution", which is different from the grid resolution and is given by the inverse of the degree of freedom. From the text I understand that having a different effective resolution for each data point along the vertical profile is a consequence of the optimal estimation approach considering entire profiles of all parameters at once (compared to the analytical solution which considers each altitude point independently). I also understand the effective resolution gets coarser where the signal is weaker and the optimal estimation gives precedence to the prior compared to the measurement. I'm not sure, however, of how to interpret a profile of effective vertical resolution as in Figures 9 or 19. In Figure 19, the effective resolution reaches > 1 km near 500m ASL, but is much finer (< 500 m) 165m above or 165m below. Is the effective resolution only to be interpreted qualitatively as an indicator of the relative importances of the signal and the prior at that particular height, or do the values themselves (500m, 1 km) mean something? If so, could you expand a bit of how to interpret them? If not, would it make sense to divide the minimum possible value (165m) by the effective resolution and provide the result as a unitless qualitative indicator of the importance of the signal for the retrieval at each height?

Yes, you are correct that the DOF of the signal is directly related to the relative importance of the signal and the prior at a given height. The inverse of that can be interpreted quantitatively as an estimate of the distance over which the state vector elements are independent and uncorrelated, at that particular point of the profile. For instance, in a region of the profile with high signal-to-noise ratio, it may be possible to retrieve the same or nearly the same number of independent state vector elements as measurements, whereas in regions with lower signal, more neighboring measurements must be considered to produce a single independent estimator of a state vector element. (To help clarify in the revised manuscript, the two sentences above have been inserted into the revision in the section "Degrees of freedom and Effective Resolution"). For the particular case of Figure 19, there are two aerosol layers with large signal-to-noise ratio separated by a very narrow clear area with low signal. If we imagine that the clear area was larger (deeper in altitude), then about 1000 m worth of data at that signal-to-noise level would need to be averaged to produce one independent estimator of aerosol extinction. Since there isn't 1000 m of clear space in the profile, the effective resolution at that particular point in the profile is a little more abstract than what would be involved in simple averaging, but it's still a useful indicator of the distance over which retrieved results are independent. The OE technique avoids having to average (or otherwise smooth) all the data to a relatively coarse

resolution over the whole profile to achieve a desired resolution at a particular point.  Yet, because the varying importance of the prior means the amount of effective smoothing varies as well, this metric is an important indicator of the consequence of that variability.

4.  Fig. 12: I'm not sure what is the point of using diamonds, squares and circles -- at their size the symbols are almost undistinguishable anyways (even in the legend).

The symbols were meant to amplify the usability for viewers with color vision deficiencies, to add another way of distinguishing the datatypes besides just color.  Your point that they were too small to distinguish is fair, so in the revision the symbols are made larger.

5.  Even though the figures are well-designed and extremely clear, 19 figures is a lot, and many have a lot of subplots. Maybe some of the figures could be omitted. I'll admit that the point of the correlation plots (6, 7, 8, 15, 16, 17) was lost on me. Maybe the authors could sum up textually whatever conclusions they drew from these figures (eg about where the uncertainties are mainly systematic) and move the figures to an appendix?

Two reviewers made similar suggestions.  In response, Figures 7 and 8 are replaced with this paragraph:

*Figure 6 shows correlation between profile quantities, but the covariance matrix also includes rows that indicate correlation between the uncertainty in the profile quantities and each the scalar quantities, K' and χ. There is significant negative correlation (not shown) between the uncertainty in the overall scaling factor, K', and the backscatter uncertainty profile, as expected. Correlations of the overall scaling factor uncertainty with lidar ratio and aerosol depolarization ratio uncertainties are near zero.  Likewise, there is predictably significant correlation between the uncertainty in the depolarization cross-talk parameter, χ, and the aerosol depolarization ratio uncertainties, whereas correlations between uncertainties in this parameter and the aerosol backscatter and lidar ratio are near zero.  All of these patterns are expected and reflect that errors (although small) in the backscatter profile are partially systematic and correlated with the calibration constant and likewise that the errors in the depolarization ratio profile are partially systematic and correlated with the depolarization cross-talk parameter.*

Figures 16 and 17 are replaced with similar but shorter summary text.  Figures 6 and 15 have been kept.  Even if the consideration of correlations in the uncertainties is a bit more esoteric than other parts of the retrieval, I believe it is important to show them as a part of due diligence.

6.  Have the authors planned to make their data analysis package (that does the optimal estimation based on lidar measurements as input) available online, for instance as a python package or something equivalent? Since the improvements are so significant compared to the more widely-used analytical approach, doing so would clearly benefit the whole lidar-based community.

We agree that that would be useful, but unfortunately no funding exists at this time to develop or maintain the code as a publicly available package in an open source language.  However, a package of code is available in the IDL programming language "as is" on request from the corresponding co-author Johnathan.W.Hair@NASA.gov.